# Evaluating the Efficacy of Electrical Vestibular Stimulation (VeNS) on Insomnia Adults: Study Protocol of a Double-Blinded, Randomized, Sham-Controlled Trial

**DOI:** 10.3390/ijerph20043577

**Published:** 2023-02-17

**Authors:** Teris Cheung, Joyce Yuen Ting Lam, Kwan Hin Fong, Calvin Pak-Wing Cheng, Alex Ho, Julie Sittlington, Yu-Tao Xiang, Tim Man Ho Li

**Affiliations:** 1School of Nursing, The Hong Kong Polytechnic University, Hong Kong SAR, China; 2The Mental Health Research Centre, The Hong Kong Polytechnic University, Hong Kong SAR, China; 3Department of Psychiatry, The University of Hong Kong, Hong Kong SAR, China; 4Integrated Services for Persons with Disabilities, Christian Family Service Centre, Hong Kong SAR, China; 5School of Biomedical Sciences, Ulster University, Coleraine BT52 1SA, UK; 6Department of Public Health and Medicinal Administration, Faculty of Health Sciences, University of Macau, Macau SAR, China; 7Department of Psychiatry, The Chinese University of Hong Kong, Hong Kong SAR, China

**Keywords:** vestibular stimulation, rct, insomnia, brain stimulation, efficacy

## Abstract

Insomnia is a common health problem in the general population. There are different ways to improve sleeping habits and quality of sleep; however, there is no clinical trial using transdermal neurostimulation to treat individuals with symptoms of insomnia in Asia. This gives us the impetus to execute the first study in Asia which aims to evaluate the efficacy of Electrical Vestibular Stimulation (VeNS) on individuals with insomnia in Hong Kong. This study proposes a two-armed, double-blinded, randomized, sham-controlled trial including the active VeNS and sham VeNS group. Both groups will be measured at baseline (T1), immediately after the intervention (T2), and at the 1-month (T3) and 3-month follow-up (T4). A total of 60 community-dwelling adults aged 18 to 60 years, with insomnia symptoms will be recruited in this study. All subjects will be computer randomized into either the active VeNS group or the sham VeNS group on a 1:1 ratio. All subjects in each group will receive twenty 30-min VeNS sessions during weekdays, which will be completed in a 4-week period. Baseline measurements and post-VeNS evaluation of the psychological outcomes (i.e., insomnia severity, sleep quality and quality of life) will also be conducted on all participants. The 1-month and 3-month follow-up period will be used to assess the short-and long-term sustainability of the VeNS intervention. For statistical analysis, a mixed model will be used to analyze the repeated measures data. Missing data will be managed by multiple imputations. The level of significance will be set to *p* < 0.05. Significance of the study: The results of this study will be used to determine whether this VeNS device can be considered as a self-help technological device to reduce the severity of insomnia in the community setting. We registered this clinical trial with the Clinical trial government, identifier: NCT04452981.

## 1. Introduction

Insomnia is one of the most common symptoms in the general population [1] and subpopulation in psychiatry, characterized by difficulty in falling asleep, maintaining sleep, and early morning awakening [2]. Insomnia can be acute, intermittent, or chronic and is often comorbid with physical and mental health conditions [3], such as decreased quality of life [4], hypertension [5], impaired immune system functioning [6], and cardiovascular disease [7]. Evaluating the symptoms of insomnia requires assessing, largely by history, whether an underlying condition can explain it. It is the diagnostic term for the symptoms of insomnia that merits specific attention. Cognitive behavioral therapy for insomnia is the preferred treatment approach because of its efficacy, safety, and durability of benefit, but pharmaceutical treatments are widely used for insomnia symptoms [3].

Insomnia is generally considered as a sleep disorder, characterized by a state of hyper arousal. Stress is believed to activate the hypothalamic–pituitary–adrenal (HPA) axis [8] and insomnia is often precipitated by stress in daily life [9]. Vestibular stimulation inhibits both the HPA axis and the sympathetic–adreno–medullar (SAM) axis and hence, decreases stress [10], excites sleep inducing areas such as the nucleus tractus solitarius (NTS) [11] and serotonergic dorsal raphe nucleus [12]. Vestibular stimulation also activates our midbrain. The activation of neurons in midbrain periaqueductal gray matter excites neurons of the rostral medulla and increases serotonin release. Activation of serotonin receptors causes the secretion of growth hormone-releasing hormone (GHRH) [13] which promotes rapid-eye movement and non-REM sleep in humans [14] which helps the modulation of sleep [15]. In addition, there is strong physiological evidence demonstrating that the vestibular system can affect REM sleep [16], and there is influence of labyrinthine inputs on the pontine reticular formation neurons involved in mediating switching between sleep states [17] and the medial vestibular nucleus has projections in regions mediating arousal, and some aspects of sleep will receive orexinergic inputs from the lateral hypothalamus [12]. Additionally, the electrical stimulation of the vestibular system can produce a rocking sensation which promotes sleep, as physical rocking is found to improve sleep in individuals with neuromuscular breathing problems [18], and the gentle swaying sensation produced by a vestibular device can significantly reduce sleep latency [19].

### 1.1. Prevalence of Insomnia in Different Countries

Insomnia co-exists with other common psychiatric disorders such as major depressive disorder, generalized anxiety disorder [20], and posttraumatic stress disorder [21]. In Hong Kong, the prevalence of insomnia in the general population was estimated at 11.9% [22] based on the presence of difficulty initiating sleep, difficulty maintaining sleep, or early-morning awakening occurring at least three nights per week in the past month. In mainland China, a study [23] found that the prevalence of any symptom of insomnia that occurred nearly every night for at least two weeks in the past year was 9.2%. In a Taiwan study, the prevalence of any insomnia symptoms that occurred either usually or all the time during the past month was 25.5% [24]. Another study showed that the prevalence of any insomnia symptoms that occurred almost every night for more than two weeks was 4.0% in Japan, 9.9% in South Korea, and 10.3% in Taiwan [25]. Although these studies have provided valuable data, owing to the differences in methodology and definition, the actual prevalence of insomnia in Asian countries remains unclear; hence, cross-cultural and cross-country comparisons are not possible [26].

### 1.2. Prevalence of Sleep Problems during COVID-19 Pandemic

The COVID-19 pandemic is an imminent public health emergency that has exacerbated or aggravated sleep disturbances in the Hong Kong community. Some researchers [27] investigated the prevalence and correlates of sleep disturbances during the early phase of the COVID-19 pandemic in Hong Kong using a web-based cross-sectional survey and convenience sampling. A total of 1138 Hong Kong adults were recruited over a two-week period in April 2020. The Insomnia Severity Index (ISI) was used to collect data on sleep disturbances, and other mood-related problems. The prevalence of insomnia (ISI score of ≥10) was 29.9%. Insufficient masks were significantly associated with worsened sleep quality; impaired sleep initiation and shortened sleep duration were significant correlates of insomnia (all *p*s < 0.05). Around 30–40% of participants reported that their overall sleep quality had worsened since the COVID-19 outbreak. Of particular note is that the prevalence of insomnia in Hong Kong has increased more than two-fold during the COVID-19 pandemic (11.9% in 2002 versus 29.9% in 2020).

### 1.3. Existing Treatment Modality for Insomnia and Past Research Findings

Medications are known to be effective in the treatment of primary insomnia in the clinical setting, but some individuals are reluctant to take the prescribed medications for fear of the drugs’ adverse effects. Psychotherapy such as cognitive behavioral therapy has also proven effective to treat insomnia but is often limited by the time and cost involved. Consequently, the majority of the individuals affected by sleeping disturbances did not seek professional help in the community in a timely fashion. Recent research findings suggested that complementary and alternative medicines (CAM) seemed to be one of the most preferred therapies for treatment of insomnia in Hong Kong, especially for the younger generations [28].

Another recent local pilot 3-armed RCT [29] examined the efficacy of integrated cognitive behavioral therapy (CBT) and acupressure in treating insomnia and its daytime impairments on 40 Chinese adults. Participants were randomly assigned to either the (1) integrated CBT with acupressure (CBTAcup) group (*n* = 14); (2) the CBT group (*n* = 13); or (3) the waitlist control (WL) group (*n* = 13). Linear mixed-effects models showed that both the CBTAcup and CBT groups had significantly lower insomnia severity (*d* = −1.74 and *d* = −2.61), dysfunctional beliefs related to sleep (*d* = −2.17 and −2.76), and mental fatigue (*d* = −1.43 and −1.60) compared with the WL group. The CBTAcup group also showed a reduction in total fatigue (*d* = −1.43) and physical fatigue (*d* = −1.45). Findings emerging from this study provided new empirical evidence to treat insomnia using integrated CBT-I and acupressure.

Insomnia is a known disorder of hyperarousal which can be attributed to increased activity in the frontal cortex during sleep. In other words, sleep disturbances have a bi-directional relationship with frontal metabolism during sleep. A recent prospective randomized controlled trial used frontal cerebral thermal therapy in 106 adults diagnosed with insomnia. Polysomnographic (PSG) was used to measure latency to persistent sleep and sleep efficiency. Results showed that two-night frontal cerebral thermal therapy improved patients’ abilities to fall asleep and had a benign safety profile [1].

### 1.4. Vestibular Stimulation

Vestibular stimulation is another alternative treatment option in the treatment and management of insomnia. A recent large-scale cross-sectional U.S. study (*n* = 20,950) found that vestibular function is closely associated with sleep quantity [30]. Results showed that 30% of individuals with vestibular vertigo reported abnormal sleep duration (15.5% short duration and 14.8% long duration). Individuals with vestibular vertigo had a higher relative risk ratio for abnormally short or long sleep duration. Further work is needed to evaluate the causal direction (s) of this association.

Modius Sleep (MS) is a non-invasive, transdermal neurostimulation device with a battery-powered headset designed to transcutaneously deliver low-level electrical current (0–1500 microamperes) to the subject’s head (neurostimulation) to treat insomnia. This MS delivers neurostimulation through two self-adhesive electrode pads which will be placed on the subject’s skin overlying each mastoid process behind the ear. MS delivers a small electrical impulse which can be (max. 1mA at 100 Hz) via Bluetooth using the mobile app to a level (0–10) where the subject can feel the tingling sensation in the skin (Table 1).

The subject will determine the level of neurostimulation once they begin to experience the gentle swaying indicating modulation of the vestibular nerve. This device will automatically turn off after 30 min of neuromodulation. The MS device will deliver a small electrical current onto the skin through the electrode pads. This electrical current serves as a neurosignaling waveform which modulates cranial nerves that influence the balance between the parasympathetic and sympathetic nervous systems. MS stimulates the head and vestibular nerves by Cranial Electrotherapy Stimulation (CES).

### 1.5. What Is Cranial Electrotherapy Stimulation (CES)? and Past Research Using CES on Insomnia 

Cranial electrotherapy stimulation (CES) has been approved by the U.S. Food and Drug Administration (FDA) for the treatment of insomnia [31]. CES is a non-invasive treatment using neuromodulation that applies alternating microcurrent transcutaneously to the human head via electrodes that can be placed on subjects’ earlobes, mastoid processes, zygomatic arches, or the maxilla–occipital junction [31]. One study [32] included 33 participants who received 4-week CES intervention, results showed that participants’ PSQI scores were significantly reduced in the post-stimulation (*p* = 0.002). More importantly, participants’ perceived level of anxiety (*p* = 0.001) and depression symptom scores (*p* = 0.024) also had a statistically significant reduction.

### 1.6. Objectives

The aim of this study is (1) to evaluate the efficacy of VeNS on insomnia severity on community-dwelling adults in Hong Kong and (2) to examine the association between VeNS data usage, insomnia severity, PSQI score, and quality of life.

### 1.7. Hypotheses

#### 1.7.1. Primary Hypothesis

Participants in the active VeNS group will have a statistically significant reduction in their (Insomnia Severity Index) (ISI) score after four weeks of treatment (T2), compared to the control participants. The effect will be maintained at 1-month (T3) and 3-months (T4) follow-up.

#### 1.7.2. Secondary Hypotheses

Higher data usage in the active VeNS group will be associated with larger improvement in insomnia severity, PSQI score, and quality of life after the 4-week treatment.

## 2. Materials and Methods

### 2.1. Trial Design

This is a two-armed, double-blind, randomized, sham-controlled trial evaluating the effects of a 4-week VeNS treatment on insomnia among Hong Kong adults in the general population. The study strictly complies with the Consolidated Standards of Reporting Trials (CONSORT) statement [33] and will be conducted in accordance with the Declaration of Helsinki [34] and Good Clinical Practice. Participants will be randomized into the active VeNS or sham VeNS group on a 1:1 ratio, balanced by their Insomnia Severity Index score, gender, and age. All the participants will be fully informed about the randomization procedures and that they have a 50% chance of receiving the active VeNS or the sham VeNS in this trial. Both groups will be measured at four time points: baseline (T1), immediately after the 4-week intervention (T2), at 1-month (T3), and 3-month follow-up (T4) (Figure 1).

### 2.2. Subjects

Samples will be recruited from our collaborative universities and NGO in Hong Kong. A flyer with registration QR code will be posted around the communal areas in the Hong Kong Polytechnic University (PolyU), University of Hong Kong (HKU), and the Chinese University of Hong Kong (CUHK). Email invitations with the QR code poster will also be sent to all staff/students and alumni across different faculties/departments at PolyU, HKU, and CUHK. A project poster will also be advertised with the School of Nursing, on PolyU Facebook, and Twitter.

### 2.3. Inclusion Criteria

To obtain a homogenous sample, the inclusion criteria will be (1) having an ISI score >15; (2) being ethnic Chinese aged 18–60 years; (3) able to understand/read Chinese; (4) currently not on prescribed or over-the-counter sleeping pills; (5) able to provide written informed consent; (6) having Wi-Fi and Bluetooth networks in iOS/Android mobile phones; (7) able to attend the face-to-face demonstration session for proper use of the MS device and return to the research assistants in the training venue (Integrative Health Clinic, School of Nursing, PolyU); (8) willing to engage with the project team on a weekly basis via WhatsApp/telephone to ensure compliance, proper usage of the MS device and report any technical issues; (9) not undergoing any extreme lifestyle changes that may impact on sleep quality (e.g., dietary changes/increase/decrease exercise levels) throughout the study period; (10) agreeing not using sleep trackers throughout the study period; and (11) not travelling to different time zones during the study period.

### 2.4. Exclusion Criteria 

Individuals with (1) a history of eczema/skin breakdown/other dermatological condition (e.g., psoriasis) affecting the skin behind the ears or having inner ear diseases; (2) HIV/AIDS infection (HIV will lead to vestibular neuropathy); and (3) use of beta-blockers/antidepressants/any other medications that may affect the neurostimulation; (4) history of stroke/epilepsy/severe head injury/neurosurgery; (5) active migraine with aura; (6) significant communicative impairments; (7) metal implant in brain or pacemaker, implanted defibrillator, deep brain stimulator, vagal nerve stimulator, etc.; (8) history of epilepsy; (9) pregnant or breastfeeding women; (10) cognitive impairment including Dementia/Alzheimer’s disease, and mild cognitive impairment; (11) history of major depressive disorder, psychotic disorder, bipolar affective disorder (depressive episode), or substance use disorders; (12) regular use of antihistamine medication in the last 6 months (because histamine receptors are present in the vestibular system); (13) history of malignancy in the past 12 months; (14) a diagnosis of myelofibrosis /myelodysplastic syndrome; (15) history of vestibular dysfunction or inner ear infections/diseases; (16) previous use of any VeNS device will be excluded.

### 2.5. Sample Size 

A recent pilot study [35] used electrical vestibular nerve stimulation on 20 adult participants with mild to moderate insomnia who were given fourteen 30-min electrical vestibular stimulation for two weeks. These sessions were delivered approximately 1 h prior to sleep onset. Results showed that participants’ ISI scores had a significant reduction after the VeNS (from mean ISI at baseline: 15.7 to mean ISI at 8.15, *p* < 0.00001).

Considering this trial is a sham-controlled trial, we hypothesize a large effect in the MS group. We used G*power version 3.1.9.7 (https://stats.oarc.ucla.edu/other/gpower/, accessed on 1 April 2022) to calculate the target sample size. With a statistical power of 95% and a statistical significance level at 0.05 to detect a large between-group effect size (Cohen’s *d*) of 0.8, each group will require 25 subjects. With an estimated attrition rate of 20% at 12-week posttreatment follow-up, we require 30 subjects per group, and hence, a total sample of 60 is required in this trial.

### 2.6. Screening and Self-Administered Questionnaire

Participants will complete a QR code online application form soliciting sociodemographic information (age, gender, educational background, marital status, monthly household income, living circumstances, employment status, psychiatric history and duration of having anxiety symptoms (in years/months), currently taking prescribed or over the counter anxiolytics (yes/no), family history of anxiety disorder (yes/no) and other psychiatric disorders, etc.) before filling in the screening tool (Insomnia Severity Index).

### 2.7. Randomization, Allocation and Masking

All consenting participants will be listed in alphabetical order according to their surnames and each participant will be assigned a unique identifier. An independent statistician will use a computer-generated list of random numbers (www.random.org, accessed on 1 June 2022) to ensure concealment of randomization. Randomization will be conducted by an independent statistician off-site using a stochastic minimization programme to balance gender, age, and ISI scores of the participants. Block randomization with blocks of 10 (total: 6 blocks) will be used to allocate treatment groups. Participants from each block will be randomly assigned to the active VeNS groups or the sham VeNS group on a 1:1 ratio. To avoid information flow, participants and research associates will be blinded to the group allocation to minimize potential contamination of the effects of VeNS or subject bias. The principal investigator will not be involved in data collection or pre-and-post VeNS measurements. Outcome measurements will be conducted by research associates who are not involved in the group allocation. Participants will be asked to guess the grouping (active VeNS vs. sham VeNS) in the face-to-face follow-up meetings after finishing the 4-week interventions, to determine the success of subject blinding [36].

### 2.8. Interventions

In this study, all participants (in both active and sham VeNS group) will collect a VeNS device for home use in our Integrative Health Clinic, SN, PolyU, after the device training delivered by the research associates. Participants will receive 20 VeNS sessions in this study, with each session lasting for 30-min over a 4-week period (i.e., Monday–Friday, total treatment time: 10 h). We believe that a four-week VeNS intervention will be sufficient enough to test the efficacy of VeNS on insomnia [35]. Participants will be followed up immediately after post-stimulation at 4 weeks and 12 weeks (Figure 1, CONSORT Flow Diagram). We believe that a post-stimulation follow-up of up to 3 months is sufficient to evaluate the sustainability of VeNS.

#### 2.8.1. VeNS group

Subjects should download the study app (Vestal) prior to each VeNS stimulation session. Subjects will put on the VeNS headset, and apply two self-adhesive electrode pads on the mastoid process behind the ear. Subjects should remain in a sitting down and resting position throughout the stimulation period. Subjects should place the electrodes on both mastoid processes simultaneously in each treatment session. Subjects will then turn on the device which will then deliver a small electrical impulse (range from 0 mA to max. 1 mA, level 0–10) at 100 Hz. Subjects should initially feel a comfortable tingling sensation around the mastoid process and soon after they should feel a gentle swaying indicating modulation of the vestibular nerve and that will be the optimal level of stimulation for individual participant. After selecting the stimulation level, the VeNS device will then deliver the stimulation for 30 min when participants press the ‘start’ button, and the device will automatically turn off. Subjects will then remain with that level until the stimulation is over.

Subjects will be able to check their activities on the study app and modify the stimulation level via the study app. The device can be paused/stopped by pressing the power button via the study app. Subjects will hear a single beep sound whenever they change the stimulation level. After the stimulation, subjects can remove the headset and dispose the electrode pads.

#### 2.8.2. Sham VeNS

Subjects in the sham-controlled group will follow the identical procedures as set in the VeNS group. Subjects will receive the initial stimulation for 30 s and then tap down to 0 mA for 20 s. Subjects in the sham-controlled group will receive a sham VeNS stimulation at a frequency of 0.8 Hz due to the fact that this frequency has a low sympathetic activation, and is less likely to cause any improvement in the sham-controlled group. An older study [37] also mentioned that sinusoidal galvanic vestibular stimulation (GVS) delivered at 0.5–0.8 Hz can cause partial entrainment of muscle sympathetic nerve activity (MSNA), and thus, we will use 0.8 Hz for the sham-controlled group in this trial. Subjects will also experience sensation of skin tingling and vestibular stimulation so that subjects in this group will believe that they have been allocated with an active device.

On completion of each VeNS session, subjects will charge the device using the micro-USB and charger provided by the project team, for usage in the next stimulation session. The device will automatically stop the stimulation after 30 min usage per day and subjects will need a 16 h lock-out period to re-use the device again. All participants will be reminded that any form of sleep tracker is prohibited in this trial.

Data will be collected via the Bluetooth for the device. The total usage per day, average intensity used, and average resistance will be logged and stored on an encrypted server. The study app will upload these data automatically when connected to Wi-Fi. All individual data with a unique trial identifier allocated at the recruitment time will be collected via REDCAP by the PI and recorded in the electronic case report form (CRF).

### 2.9. Fidelity

To ensure the fidelity of the intervention, the project team will ascertain whether the interventions will be delivered as designed. To ensure compliance, the project team will monitor the duration of usage data on a weekly basis (Friday). If any subject’s weekly usage is <2.5 h, the research associates will send a WhatsApp text message as a gentle reminder and investigate if subject experiences any technical issues in using the device. The PI will work closely with the research personnel to optimize compliance throughout the study period.

### 2.10. Safety, Adverse Effects and Risk Indicators

Past research [38] was conducted on skin inspection of the mastoid area, otoscope examination of the inner ear, and formal audiometry testing on 25 VeNS long-term daily or regular users (2–3 times/week) examining the safety of the MS device. Results showed that these long-term NeVS users had a mean duration of VeNS use for 22 months and they did not show any abnormality through examination of the mastoid areas, ear canal, or tympanic membranes nor audiogram assessments. In this regard, VeNS is safe and well tolerated by Modius Sleep users.

However, some subjects may have skin irritation at the electrodes’ sites, sensation of disequilibrium, nausea/vomiting, headache, and an electrical tingling sensation. Hence, an adverse events checklist will be used to monitor/quantify the occurrence of these events between the VeNS group and the sham VeNS group. Subjects are encouraged to contact the project team members should these adverse effects become too unmanageable. The PI will then assess the subjects’ physical and mental conditions and decide whether they could continue the intervention or not.

### 2.11. Ethical and Data Security Considerations

Ethical approval will be obtained from the Human Subjects Ethics Sub-committee, Hong Kong Polytechnic University. This study adheres strictly to the Declaration of Helsinki ethical principles developed by the World Medical Association. All participants will be covered by trial insurance. Potential risks involved in NeVS will be clearly indicated in the Information Sheet. Voluntary participation, anonymity and confidentiality and the right to withdraw will be respected.

Participants’ data in both groups will be stored in two separate datasets with an identifier linking these data. Both sets of data will be encrypted using TrueCrypt (http://www.truecrypt.org). The data from baseline, and the four-week, one-month and three-month follow-up will be linked according to personal data. All precautions in data protection will be taken, as suggested by TrueCrypt. To prevent leakage of personal data, only the PI will have access to the personal dataset. Written consent will be obtained from all participants. An information sheet containing the purpose of this trial and the VeNS procedures and the Modius Stress leaflet will be provided to all participants. Participants will be informed of their anonymity; withdrawal or non-compliance will not result in any consequences.

### 2.12. Outcome Evaluation (Primary and Secondary Outcomes)

The primary objective of this study is to evaluate the effects of VeNS on participants’ insomnia severity among adults in Hong Kong. Secondary objectives include examining the effects of VeNS data usage on insomnia severity, sleep quality, and quality of life.

#### 2.12.1. Primary Outcome

##### Insomnia Severity

Insomnia severity will be assessed by the Chinese version of the Insomnia Severity Index (ISI) which consists of 7 items measuring day and night symptoms of insomnia on individuals. This ISI consists of seven items (1: perceived difficulty, 2: falling asleep; 3: time of awakening; 4: satisfaction with current sleep pattern; 5: interference with daily functioning; 6: noticeability of others of impact of lack of sleep; 7: degree of perceived distress/concern caused by the sleep problem). Subjects will rate each question on a 5-point Likert scale (0–4), with score ranges of 0–28, with 15–21 indicating moderately severe insomnia. ISI has been used in the Chinese population with good psychometric properties, with Cronbach’s alpha 0.81 and item-to-total correlations in the range of 0.34–0.67 [39].

#### 2.12.2. Secondary Outcomes

##### Sleep Quality

The Pittsburgh Sleep Quality Index (PSQI) [40] will be used to assess sleep quality over the past month. The PSQI consists of 19 items in total. Participants type in their responses to the first 5 questions asking about their bedtime, time taken to fall asleep, wake up time, actual sleep time, and time in bed in the past month. Next, they respond to the remaining 14 questions that ask about the frequency of sleep problems, use of sleeping pills to go to sleep and daytime sleepiness on a Likert-type scale ranging from 0 to 3, with a higher value indicating more severe sleep problem. A score of ≥5 indicates poor sleep quality. Total score ranges are 0–21. If total score increases, it indicates a decrease in sleep quality. The Chinese version of the PSQI is a reliable and valid instrument to assess sleep quality among the Hong Kong Chinese population [41].

##### Quality of Life

Quality of life will be assessed by the Chinese version of the 36-item Short Form Health Survey (SF-36) [42]. The SF-36 includes 36 questions related to an individual’s QoL on eight scales: physical functioning (PF), role—physical (RP), bodily pain (BP), general health perceptions (GH), vitality (VT), social functioning (SF), role—emotional (RE), and mental health (MH). The raw scores for each scale are transformed to a scale of 0–100, with higher scores indicating ‘better’ QoL. SF-36 is summarized in two component summary scores: the Physical Component Summary (PCS) and the Mental Component Summary (MCS). This Chinese version of SF-36 is a valid and reliable instrument, with the overall Cronbach’s α coefficient of 0.943, while the Cronbach’s α coefficients for each of the dimensions found to be all >0.70 [43].

All the primary and secondary outcomes will be assessed at baseline, at 4-week posttreatment, 1-month, and 3-month posttreatment for immediate, and short-term effects [44].

**Trial Status:** This trial was registered with ClinicalTrials.gov on 9 August 2022 (protocol version) (identifier: NCT04452981). Recruitment commenced between 1 June 2022 and 30 December 2022.

## 3. Statistical Analyses

All statistical analyses will be performed by R for Windows (R version 4.1.0). Normal distribution of the data will be assessed by a Q-Q plot. Descriptive analysis that include means and standard deviations (SD) for the continuous variables will be presented, while numbers and percentages for the categorical variables will also be shown. A *p* value < 0.05 is considered statistically significant. Sociodemographic differences between the active VeNS group and the sham VeNS group will be analyzed using Chi-square test and *t* test. Should there be significant differences in sociodemographic factors between the two groups, covariates will be considered as confounding variables in the statistical analyses. Normality of the primary outcome (ISI) scores will be tested by the Shapiro–Wilk test. The independent samples *t* test will be used to test the baseline difference between the two groups.

A mixed model will be used to test the group (between-subject factor), time (within-subject factor), and group × time interaction effects of the ISI score between the active VeNS group and the sham VeNS group. Post hoc comparisons between groups and time points will be conducted using a *t* test with Bonferroni correction. A Cohen’s *d* effect size for each outcome will be calculated, where *d* = 0.2, 0.5, and 0.8 correspond to small, medium, and large effect size. Participants’ total usage time will also be used to investigate the effectiveness of the VeNS on anxiety and total usage time will be used as a continuous covariate in a linear regression model for ISI total score. Missing data will be managed by multiple imputation [45] by adopting the last observation carried forward (LOCF) method.

## 4. Significance/Novelty of This Study

This is the first double-blind, randomized, sham-controlled trial evaluating the efficacy of the VeNS device on the reduction of insomnia symptoms in Asia, and this is also a multi-site comparison study with the U.K. and India. Mental health promotion in primary care is pivotal to reduce the global prevalence of primary insomnia and other sleep-related problems. If our findings emerging from this study can prove that the VeNS device can help reduce insomnia problems, it would bring about a major breakthrough in neuroscience research and cast a significant research and community impact to the general population. Community-dwelling individuals can have a more user-friendly but safe treatment option to ease their sleep problems, which in the long run can improve their quality of life, and biological and mental wellbeing. From stakeholders and health policymakers’ perspectives, improving mental health in the general populations will also help reduce the global disease burden.

## 5. Conclusions

This is a collaborative study with the University of Ulster in the U.K. Findings emerging from a cross-cultural context may shed new insight on the efficacy of VeNS and add new knowledge to the treatment of insomnia in the East and West. More importantly, findings from this study can provide scientific evidence to affirm whether this VeNS device can be considered as a self-help technological device to reduce primary insomnia in the community setting. If vestibular stimulation can be proven to reduce the symptoms of insomnia in this study, individuals may have another self-help treatment option to manage their sleep disturbances at the primary care level and foster their mental health on a short- and long-term basis. Taking proactive steps in the self-management of early psychological symptoms are critical milestones in contributing to the success of any treatment strategies. Given the increased global prevalence of insomnia in Hong Kong and nationwide, this NeVS device may help mitigate the detrimental impact brought by insomnia in a timely fashion. Increased awareness into mental problems may also help reduce the global disease burden and psychiatric morbidity in the long run.

## Figures and Tables

**Figure 1 ijerph-20-03577-f001:**
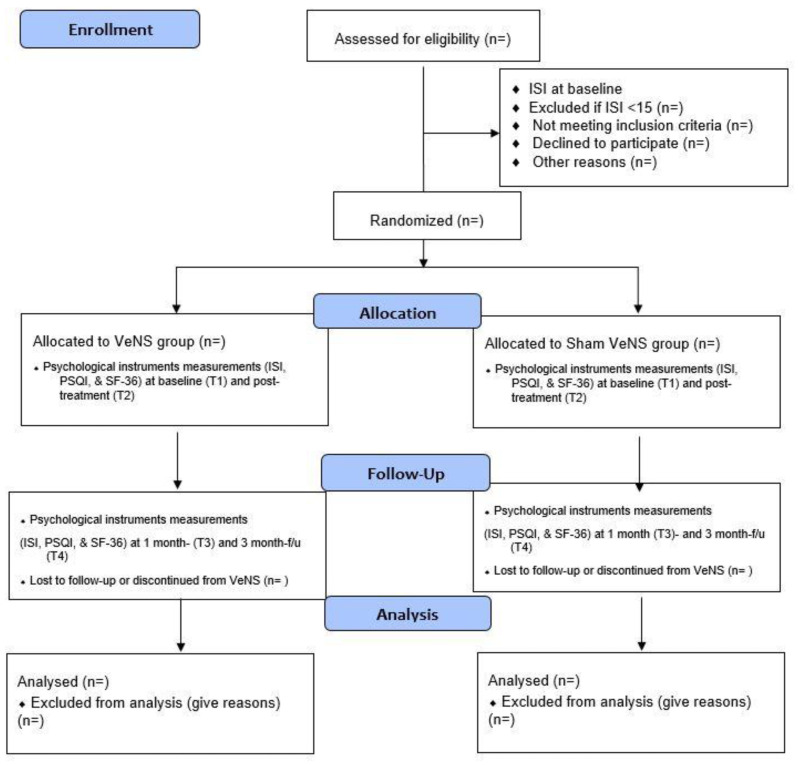
Flow diagram for study design.

**Table 1 ijerph-20-03577-t001:** The electrical current of each level.

Level	Stimulation Level
0	No stimulation applied (0 mA)
1–3	Stimulation applied (0.5 mA)
4–7	Stimulation applied (0.7 mA)
8–10	Stimulation applied (1 mA)

## Data Availability

The original contributions presented in the study are included in the article. Further inquiries can be directed to the corresponding author.

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
