# Peer review of "Evaluating the Efficacy of Electrical Vestibular Stimulation (VeNS) on Insomnia Adults: Study Protocol of a Double-Blinded, Randomized, Sham-Controlled Trial"

_ijerph, 2023, doi:10.3390/ijerph20043577_

Round 1

Reviewer 2 Report

A practical study of the effectiveness of vestibular stimulation on the severity of insomnia in the general population.

The study protocol is well presented, easy to follow, consistently illustrated.

 Some concerns:

 Why is the term vestibular stimulation used /because of the placement/ when it is electrical stimulation in itself?

In section 2.6. Screening and self-administered questionnaire

If the relationship between vestibular function and sleep is discussed, in this sense it can be said that with other subclinical mental symptoms such as depression and anxiety, which fall outside the specified exclusion criteria, accompanying insomnia can be observed.

In this sense, how should the effect of electrical /vestibular/ stimulation conducted in this way be considered on sleep - as a primary or as a secondary in relation to the influence of these subclinical mental phenomena.

It is appropriate to list the scales that have been used to exclude the relationship between insomnia and these subclinical symptoms.

 Reviewer

Round 2
